# Targeting Cellular Senescence to Enhance Human Endometrial Stromal Cell Decidualization and Inhibit Their Migration

**DOI:** 10.3390/biom15060873

**Published:** 2025-06-16

**Authors:** Julia Delenko, Nathaniel Hyman, Prodyot K. Chatterjee, Polona Safaric Tepes, Andrew J. Shih, Xiangying Xue, Jane Gurney, Andrew G. Baker, Cheng Wei, Daniel Munoz Espin, Ljiljana Fruk, Peter K. Gregersen, Christine N. Metz

**Affiliations:** 1Northwell, 2000 Marcus Ave, Suite 300, New Hyde Park, NY 11042, USA; julgwen@gmail.com (J.D.); nhyman@northwell.edu (N.H.); pchatter@northwell.edu (P.K.C.); psafartepes@northwell.edu (P.S.T.); ashih@northwell.edu (A.J.S.); xxue@northwell.edu (X.X.); janengurney@gmail.com (J.G.); 2The Feinstein Institutes for Medical Research/Northwell, 350 Community Drive, Manhasset, NY 11030, USA; 3The Donald and Barbara Zucker School of Medicine at Hofstra/Northwell, 500 Hofstra Blvd, Hempstead, NY 11549, USA; 4Early Cancer Institute, Department of Oncology, University of Cambridge, Hills Road, Cambridge CB2 0XZ, UK; agb59@cam.ac.uk; 5Department of Chemical Engineering and Biotechnology, University of Cambridge, Philippa Fawcett Drive, Cambridge CB3 0AS, UK; cw796@cam.uk (C.W.); dm742@cam.ac.uk (D.M.E.); lf389@cam.ac.uk (L.F.)

**Keywords:** endometriosis, flavonoids, infertility, menstrual effluent, senescence, senolytic, senomorphic, quercetin

## Abstract

Cellular senescence leads to stable cell cycle arrest and an inflammatory senescence-associated secretory phenotype that varies with stressor and cell type. To mitigate these effects and improve health, senotherapeutics (e.g., senolytics and senomorphics) have been developed. Senescent-like endometrial stromal cells (eSCs) lining the uterus of patients with endometriosis and infertility are proposed to impair decidualization, a differentiation process required for uterine receptivity in humans. Quercetin, a natural flavonoid senolytic, dramatically improves decidualization and reduces endometriosis in rodent models. However, little is known about the comparative effects of various senotherapeutics on eSCs. Using menstrual effluent-derived eSCs, we evaluated the effects of flavonoid and non-flavonoid compounds on eSC functions associated with endometriosis, aiming to identify optimal senotherapeutics for future clinical trials. Among flavonoids tested, all senolytics (quercetin, fisetin, and luteolin) and kaempferol, a senomorphic, significantly improved decidualization without cytotoxicity. Although non-flavonoids exhibited notable cytotoxicity, dasatinib, but neither ABT-737 nor navitoclax, enhanced decidualization. Flavonoid senotherapeutics and dasatinib significantly inhibited eSC migration. Mechanistic studies revealed that all flavonoids and dasatinib suppressed AKT phosphorylation and upregulated p53 expression. Notably, only quercetin and fisetin reduced ERK1/2 phosphorylation. Furthermore, flavonoid-senolytics and dasatinib consistently eliminated senescent eSCs. These findings support future studies to assess the therapeutic potential of in vivo supplementation with flavonoid senolytics on eSC function using menstrual effluent.

## 1. Introduction

Senescent cells exhibit characteristic features including stable growth arrest and a senescence-associated phenotype (SASP) defined by increased production of inflammatory mediators and extracellular matrix proteins, altered cell metabolism, and resistance to cell death [1]. Although most often associated with aging-related conditions, senescent cells can be found in non-aged organisms because of replicative stress, inflammation, infections, and/or oxidative stress [2,3]. Numerous senotherapeutics, including senomorphics (that suppress SASP factor production and inflammation) and senolytics (that eliminate senescent cells via senolysis) have been developed to target senescent cells and have been tested in numerous pre-clinical and clinical studies [4,5]. Specifically targeting senescent cells with senolytics, namely quercetin (a natural occurring flavonoid) and dasatinib (a kinase inhibitor), has been shown to reduce senescent cell burden, improve physical well-being, and prolong lifespan in mice [6] and has shown success in clinical trials for treating pulmonary and renal fibrosis in humans [4,5,7].

The potential effects of cellular senescence on uterine health, including fertility and endometriosis, have been described [8,9,10,11]. Endometriosis is a complex and chronic condition affecting approximately 10% of females of reproductive age and is often accompanied by infertility. It is characterized by the growth of lesions containing endometrial-like stromal cells and glandular epithelial cells, mainly in the peritoneal cavity. Chronic pelvic pain ranging from mild to severe, and other symptoms along with inadequate treatment options, negatively affect patients’ participation in social and athletic activities, their productivity, and overall life quality, and add emotional, societal, and financial burdens. Numerous studies have reported alterations in the eutopic endometrium (lining the inside of the uterus) in patients with endometriosis—namely, enhanced inflammation [12,13,14,15]. Menstrual tissues (containing shed endometrium) are delivered to the pelvic cavity via retrograde menstruation via the fallopian tubes in almost all menstruators [16] and this process, among others, has been implicated in endometriosis for almost 100 years [17]. Single cell RNA sequencing (scRNAseq) of shed endometrial tissues in menstrual effluent (ME) has provided detailed analysis of immune and non-immune cells, including endometrial stromal cells (eSCs), that are delivered to the peritoneal cavity during menstruation. Compared with unaffected controls, eSCs from endometriosis patients exhibit aberrant eSC decidualization markers [9]. Decidualization, a progesterone driven differentiation process during the mid-secretory phase, is essential for embryo implantation in humans [18] that can be negatively impacted by inflammatory mediators [19,20]. Furthermore, scRNAseq analysis of ME reveals elevated numbers of pro-inflammatory and senescent-like eSCs [9] that likely resist apoptosis. These scRNAseq findings are consistent with reported alterations in the eutopic endometrium of patients with endometriosis [9,12,14,15] and impaired eSC decidualization observed in patients with endometriosis [19,21,22,23], as well as infertility, and other conditions [20,24,25]. Together, these findings support a model in which senescent cells compromise eSC function within the uterus, potentially contributing to infertility and the formation and progression of endometriosis lesions. Given that impaired decidualization has been linked to endometriosis, infertility, and other reproductive disorders, targeting this process may offer a promising therapeutic strategy.

Currently, most pharmacologic therapies for endometriosis are hormone-based and target pain; they likely do not halt disease progression, cure disease, or prevent disease, and many are ineffective and/or poorly tolerated [26,27]. Given that elevated levels of senescent-like eSCs in ME have been reported in patients with endometriosis [9], it is not surprising that the natural flavonoid senolytic quercetin showed therapeutic potential in rodent models of endometriosis and on eSC function in vitro [28,29,30,31]. Rodent models of endometriosis revealed that quercetin treatment significantly reduced the size of autoimplanted uterine tissues and reduced markers of inflammation [28,29]. Importantly, quercetin not only enhanced decidualization by control-eSCs, but it also corrected impaired decidualization responses in eSCs from endometriosis patients to levels observed with control-eSCs [31]. As one of numerous flavonoid-based senotherapeutics, quercetin exhibits diverse biological activities and has shown promising results across numerous pre-clinical and clinical studies [32,33,34].

Given the variability in cellular pathways and the selective actions of senotherapeutics based on distinct senescent cell fates [4], we systematically compared a panel of well-studied senotherapeutics for their effects on eSCs. Specifically, we evaluated senolytics and senomorphics, including natural flavonoids (quercetin, fisetin, kaempferol, and luteolin) and synthetic non-flavonoids (ABT-737, dasatinib, and navitoclax) (Table 1) for their effects on key eSC functions, decidualization, migration, and cytotoxicity, to evaluate their potential therapeutic relevance for patients with endometriosis and related infertility. Additionally, we examined underlying mechanisms by which these various senotherapeutics influence eSC function.

## 2. Materials and Methods

### 2.1. Chemical and Reagents

Quercetin was purchased from Alfa Aesar/Fisher Scientific (Waltham, MA, USA); 8-bromoadenosine 3′,5′-cyclic monophosphate sodium salt (cAMP), and medroxyprogesterone acetate (MPA) were purchased from Sigma-Aldrich (St. Louis, MO, USA). ABT-737, dasatinib, fisetin, kaempferol, luteolin, and navitoclax were purchased from MedChemExpress (Monmouth Junction, NJ, USA). All senotherapeutics, including quercetin, were prepared in DMSO, as directed by the manufacturer and stocks were stored at −80 °C (for up to 3 months). See Table 1 for details on the senotherapeutics. Palbociclib was purchased from MedChemExpress and prepared as described above.

### 2.2. Isolation and Culture of Menstrual Effluent-Derived Endometrial Stromal Cells (eSCs)

Endometrial stromal cells (eSCs) were isolated from menstrual effluent (ME) collected from participants enrolled in the ROSE study following informed consent (IRB #13-376A). All participants were females of reproductive age (18 to 45 years old) who were not pregnant, not breastfeeding, not using hormonal contraceptives, and who were currently menstruating and willing to provide menstrual effluent (ME). Healthy, unaffected participants (controls) self-reported no symptoms or diagnosis of endometriosis, polycystic ovary syndrome (PCOS), or adenomyosis. Patients with endometriosis had a positive pathology report (following diagnostic laparoscopy) and reported no history of PCOS or adenomyosis.

ME was collected using menstrual cups (donated by Diva International, Toronto, Ontario, Canada) and processed to grow endometrial stromal cells (eSCs), as previously described [19,31]. Briefly, ME-derived eSCs were cultured/expanded in DMEM containing 10% mesenchymal stem cell-fetal bovine serum (FBS), 1% penicillin-streptomycin-glutamine (PSQ) (Gibco/Thermo Fisher, Waltham, MA, USA), and normocin (1:500) (Invivogen, San Diego, CA, USA) (maintenance media) at 37 °C with 5% CO_2_ [19,21,31]. Passage 3-4 (p3-4) eSCs were used for experiments (unless indicated). Sample sizes for cell-based assays were consistent with prior studies [19,21,31], as the study of ME-eSCs does not involve ethical, time, or cost issues that warrant sample size calculations [35].

### 2.3. Decidualization Assays

Confluent monolayers of ME-derived eSCs (p3-4) from healthy controls were plated in 96-well plates at 2.5–3 × 10^4^/well. Four to eight hours later, media was aspirated and replaced with assay media (DMEM, 2% FBS, PSQ, and normocin). The next day, cells were treated with either vehicle (assay media containing ≤0.1%DMSO (final), equivalent to the highest dose of DMSO used when treating with various agents), quercetin (25 μM), or various senotherapeutics (flavonoids: 25–50 μM; non-flavonoids: 50–250 nM) (all prepared in DMSO, as described above) diluted in assay media for 4 h before the addition of cAMP (0.5 mM) + MPA (10^−7^ M) to induce decidualization (3–4 wells/condition), followed by measurement of IGFBP1 protein 48 h later, as previously described [19,31]. Specifically, doses of deciduogenic agents and time points of assessment were based on our prior studies and previously published studies, allowing for the comparison of results across studies. Additionally, the 48 h time point is consistent with our prior studies using ME-derived eSCs where the decidualization response is robust enough to detect both positive and negative effects and prior to its peak or completion [19,21,31]. Furthermore, the combination of deciduogenic stimuli (cAMP + MPA) was chosen because recent transcriptomic and cell-based studies suggest it best reflects decidualization in vivo [36]. Note: concentrations of flavonoids used were guided by our previous studies with quercetin [31] (using doses that inhibited proliferation and enhanced decidualization without cytototoxicity) and by published studies (quercetin [37,38,39], fisetin [40,41,42]; kaempferol [37,43,44]; and luteolin [45,46]) that used similar concentrations. Concentrations for non-flavonoids were based on a study using dasatinib-treated human cells (100 nM) [47] and initial studies; doses for non-flavonoids were mainly below those described in the literature for primary cells due to cytotoxicity.

### 2.4. Cytotoxicity Assays

As described above, confluent monolayers of control ME-derived eSCs (p3-4) in assay media were treated with vehicle or various senotherapeutics (flavonoids: 25–50 μM; non-flavonoids: 50–250 nM) (n = 4/condition). After 48 h, cytotoxicity was assessed using neutral red viability assays, as previously described [31,48].

### 2.5. Western Blotting

Endometrial stromal cells (eSCs, p2-4) from healthy controls and endometriosis patients were plated at 2–4 × 10^5^ cells/well in maintenance media in a 6-well plate (2 wells/condition). When confluent, maintenance media was replaced with assay media. The next day, eSCs were treated with vehicle, quercetin (25 μM), fisetin (25 μM), kaempferol (25 μM), luteolin (25 μM), ABT-737 (100 nM), dasatinib (100 nM), or navitoclax (100 nM) for 4 h. Cells were washed with PBS and lysed with RIPA lysis buffer containing protease and phosphatase inhibitors (Halt™ Protease and Phosphatase Inhibitor Cocktail, Thermo Fisher) and analyzed by western blotting, as in [31]. Briefly, SeeBlue Plus2 pre-stained molecular weight standard (Thermo Fisher) or lysates (20–45 µg protein/lane) were separated by electrophoresis, transferred onto Immobilon FL-PVDF membranes, and immunoblotted with primary antibodies: phospho-AKT (S473) monoclonal, AKT polyclonal, phospho-PRAS40 (T246) monoclonal, phospho-ERK1 (T202/Y204), phospho-ERK2 (Thr185/Tyr187) monoclonal, ERK1/2 polyclonal, p53 polyclonal, and GAPDH monoclonal (1:1000) (Cell Signaling Technology, Danvers, MA, USA), as per manufacturer’s instructions. Band densities were quantified using NIH Image J, version 1.x and normalized to GAPDH protein levels and to specific total protein where applicable (e.g., p-AKT:AKT) on the same blots. Uncropped blots and band densities for control-eSCs are shown in Appendix A, respectively. Uncropped blots and band densities for endometriosis-eSCs are shown in Appendix A.

### 2.6. Single Cell RNA Sequencing (scRNAseq) of Quercetin vs. Vehicle-Treated eSCs

Control-eSCs (p1, from 4 participants) were plated in maintenance media at 7 × 10^5^ cells/well in 6-well plates. Media was replaced with assay media and the following day eSCs were treated with either vehicle or quercetin (25 μM) (2 wells/condition). After 4 h, cells were harvested and processed for single cell RNA sequencing (scRNAseq) using 10× Chromium single cell 3′ v3 RNA Profiling preparation (10× Genomics, Pleasanton, CA, USA). Samples were sequenced to at least 25,000 mean reads per cell (at MedGenome, Foster City, CA, USA). Fastq files were demultiplexed, aligned to the human reference genome GRCh38 using 10× cellranger 7.0.0. Downstream analysis was carried out using Seurat. Cells were filtered out if they had >25% mitochondrial reads, <200 UMIs (unique molecular identities) or >9000 UMIs per cell. SCTransform was used to normalize gene expression and harmony was used to remove person-specific and batch-specific effects. Cell cycle phase was assessed in Seurat and actively cycling cells were removed from consideration, with 9420 cells that passed all QC metrics. Differentially expressed genes between quercetin- vs. vehicle-treated were analyzed using a Wilcoxon rank test. scRNAseq data are included in Appendix A. All datasets were deposited in the National Center for Biotechnology Information/Gene Expression Omnibus (GEO) accession number (GSE299492).

### 2.7. Migration Assays

Migration assays were performed with control ME-eSCs using the IncuCyte^®^ SX5 imaging platform (Sartorius, Ann Arbor, MI, USA), according to the manufacturer’s guidelines for wound-migration assays. Briefly, IncuCyte^®^ Cell Migration 96-well plates (Sartorius) were coated with bovine collagen I (50 µg/mL, Corning, Glendale, AZ, USA). After a PBS wash, eSCs (p3-4) were plated at 2 × 10^4^ cells/well in maintenance media. Cells were treated with vehicle or senotherapeutics (flavonoids: 25 μM; non-flavonoids: 100 nM) 24 h later. After an overnight incubation (when confluent), uniform cell-free zones were created in each well using the Incucyte^®^ WoundMaker 96-pin tool. Plates were washed with DMEM to remove excised cells and maintenance media containing mitomycin C (2 µg/mL, to prevent cell proliferation) and either vehicle or various senotherapeutics (n = 4 wells/condition per subject; flavonoids: 25 μM; non-flavonoids: 100 nM). Plates were incubated in the IncuCyte^®^ (at 37 °C/5% CO_2_) and read every 2 h for at least 48 h. For comparing senotherapeutics, wells were analyzed when vehicle-treated wells for each subject’s eSCs reached about 50% closure (on average). Data are presented as % wound closure compared to vehicle-treated cells.

### 2.8. Assessment of Senescence Phenotype

Senescence phenotype assessed by NanoJagg accumulation: Cellular senescence and the elimination of senescent cells were analyzed using NanoJaggs, fluorescent probes preferentially taken up by senescent cells and stored in lysosomes, as described by Baker et al. [49]. To produce senescent cells, eSCs (p6-8) in maintenance media were treated with palbociclib (PALB, 2.5 µM) three times per week during the first week, followed by once per week for about 2 ½-3 weeks. Also, paired vehicle-treated eSCs (from each participant) were cultured in maintenance media and passaged once or twice weekly, as needed. PALB-eSCs (senescent) did not appear to proliferate and had a portion of their media replaced weekly. Paired vehicle-treated vs. PALB-eSCs (from the same subject) were plated in 96-well plates (1–2 × 10^4^/well) and once attached, treated with vehicle or senotherapeutics (flavonoids at 25 µM, non-flavonoids at 100 nM) (3–4 wells/condition). The next afternoon, media was replaced with 0.2%FBS-media containing vehicle or various senotherapeutics (flavonoids at 25 µM, non-flavonoids at 100 nM) and the next morning NanoJaggs (20 µg/mL) were added. After 6 h in the dark at 37 °C/5% CO_2_, eSCs were washed with PBS and lysed in water following two freeze/thaw −80 °C-RT) cycles. Cell lysates were transferred to black plates and NanoJagg uptake was measured at 633 nm Ex/720 nm Em in a TECAN Spark^®^ (Morrisville, NC, USA) multimode microplate reader. Data are presented as NanoJagg uptake corrected for the number of cells per well (after trypsinization and counting cells from untreated wells). Senescence was confirmed by cell cycle arrest and senescent biomarker expression (see below).

Senescence phenotype assessed by growth arrest: Briefly, paired vehicle-treated and PALB-eSCs (p6-8) from the same subjects (n = 4–6 wells per condition) were plated at 1–2 × 10^3^ cells/well in 96-well plates in maintenance media and allowed to proliferate at 37 °C/5% CO_2_. After 96 h, cells were washed with PBS and assayed using the CyQUANT™ Cell Proliferation Assay (Thermo Fisher), as described [31].

Senescence phenotype assessed by the expression of senescence biomarkers: As described above, vehicle- and PALB-eSCs were plated in maintenance media in a 6-well plate and treated with vehicle or quercetin (25 μM) (2 wells/condition). The next day, media was replaced with assay media and cells were treated again with vehicle or quercetin (25 μM) for 8 h. Cell lysates were processed as described above for western blotting, except membranes, which were immunoblotted with primary antibodies for p16 and GAPDH (Cell Signaling Technology), and phospho-Rb (Ser807/Ser811) (Proteintech, Rosemont, IL, USA). Uncropped blots and band densities are shown in Appendix A. Additionally, we sorted NanoJagg-positive and NanoJagg-negative eSCs and assessed senescence biomarkers. Briefly, PALB- or vehicle-treated eSCs (in 100 mm plates) were incubated as described above with NanoJaggs (20 µg/mL) for 8 h at 37 °C/5% CO_2_. Then, eSCs were washed with HBSS and incubated with DAPI (300 nM) in HBSS at RT for 10 min. After washing, live NanoJagg-positive (senescent) and live NanoJagg-negative (non-senescent) eSCs were isolated using BD FACSAria II Cell Sorter (BD Biosciences, San Jose, CA, USA) with a 100 μm nozzle using a doublet discrimination strategy. Live (DAPI-negative) NanoJaggs-positive cells were excited and detected in the APC-Cy7 emission spectrum. Gates for NanoJagg-positive and NanoJagg-negative cell populations were defined based on positive and negative control-eSCs. Sorted eSCs were collected, lysed, and assessed for senescence markers p-Rb and p16, and GAPDH (as described above). Full blots and band densities are shown in Appendix A.

### 2.9. Statistical Analyses

Analyses and graphical presentations were performed using GraphPad Prism 10.4.1 software. For data with two groups, groups were compared using the Wilcoxon matched-pairs signed-rank test (because data did not meet criteria for parametric testing: normal distribution and homogeneity of variances). For data with multiple groups, data were analyzed using the Kruskal–Wallis (one-way ANOVA on ranks) test, with appropriate post hoc testing. *p*-values less than 0.05 were considered significant.

### 2.10. Patents

CNM and PKG are co-inventors of a US patent entitled “Methods for detecting and treating endometriosis (using menstrual effluent) (US 12,146,886B2)”. Currently, they receive no royalites or financial benefits related to this patent. L.F., D.M.E., and A.G.B. are co-inventors of NanoJaggs, covered by a patent entitled “ICG nanoparticles for senescence imaging (PCT/GB2023/050386)”.

## 3. Results

### 3.1. Senotherapeutic Agents Have Differential Effects on eSC Decidualization

Several flavonoid senotherapeutics were tested, including fisetin, luteolin, and kaempferol. Both fisetin and kaempferol (flavonols) significantly increased eSC decidualization at 25 and 50 µM when compared to vehicle treatment (Figure 1A,B), while luteolin (flavone) increased eSC decidualization only at 25 µM compared to vehicle treatment (Figure 1C). Decidualization was assessed by measuring IGFBP1 levels, normalized to vehicle. When flavonoids, including quercetin, were compared at 25 µM, all flavonoids enhanced eSC decidualization based on IGFBP1 production (Figure 1D). Both kaempferol and luteolin induced decidualization similarly to quercetin, while fisetin appeared less potent (Figure 1D).

Three synthetic non-flavonoid senotherapeutics, ABT-737, dasatinib, and navitoclax, were tested at various concentrations (50–250 nM) for effects on eSC decidualization (Figure 1E–H). Dasatinib, a broad-spectrum tyrosine kinase inhibitor, significantly increased eSC decidualization (Figure 1F,H), while BH3 mimetics ABT-737 (Figure 1E,H) and navitoclax (Figure 1G,H) significantly reduced decidualization when compared to vehicle treatment.

### 3.2. Some Senotherapeutics Are Cytotoxic

Given the differential effects of studied senotherapeutics on IGFBP1 production by decidualizing eSCs, we next assessed their cytotoxicity. While fisetin and kaempferol were not cytotoxic (Figure 2A,B), high-dose luteolin (50 µM), which did not significantly enhance IGFBP1 production (Figure 1C), exhibited mild cytotoxicity (Figure 2C). At 25 µM, all flavonoids, except quercetin, demonstrated significant cytoprotective effects relative to vehicle-treated control-eSCs (Figure 2D). All subsequent experiments involving flavonoids were conducted using 25 µM.

In contrast, the non-flavonoid senotherapeutics displayed cytotoxicity at concentrations tested (50–250 nM) (Figure 2E,H). Dasatinib was the least cytotoxic (Figure 2F), even at 50–100 nM doses that significantly enhanced decidualization (Figure 1F,H), when compared to ABT-737 and navitoclax at the same doses (Figure 2E,G,H). Based on these findings, we selected 100 nM for all subsequent non-flavonoid experiments, as this previously published dose offered a favorable balance of efficacy and minimal cytotoxicity, particularly in the case of dasatinib (Figure 1F,H).

### 3.3. Quercetin-Treated eSCs Upregulate Gene Expression Related to Cell Migration and Cell Survival

Based on our prior data [26], single cell RNA-sequencing (scRNAseq) analysis was performed using eSCs treated with vehicle vs. quercetin (25 µM). Quercetin treatment upregulated and downregulated numerous genes (Supplemental Data S4). Closer examination of the data identified numerous genes associated with cell survival (e.g., *AEBP2*, *CITED2*, *GSTO2*, *WNT2B*, *DKK1*, and *MSH4*) and cell migration (e.g., *EZR*, *ZEB1*, *MEIS2*, *PRR16*, and *DGKI*) (Figure 3).

### 3.4. Multiple Senotherapeutics Effectively Inhibit Cell Migration

Building on the scRNAseq results (Figure 3) and prior studies revealing that senolytics affect cell migration and invasion [50,51,52,53], we assessed our senotherapeutic panel for effects on eSC migration using the wound closure assay. As shown in Figure 4A–C, quercetin significantly reduced eSC migration at 12, 18, and 24 h post-wounding, though the rate of migration varied across participants’ eSCs. All tested flavonoid senotherapeutics significantly inhibited eSC migration (Figure 4D), with the senomorphic kaempferol being the least effective. Among the non-flavonoid senolytics, only dasatinib significantly suppressed eSC migration (Figure 4E), while the BH3 mimetics ABT-737 and navitoclax, at equivalent doses, showed no significant effect (Figure 4E).

### 3.5. Effects of Senotherapeutics on eSC Signaling Pathways

Next, we evaluated the effects of these senotherapeutics on signaling pathways implicated in eSC decidualization, previously shown to be regulated by quercetin [31]. As shown in representative western blots from one control’s eSCs (Figure 5A), several senotherapeutics (mainly flavonoids) altered the expression of multiple signaling proteins when compared to vehicle treatment. The effects of the senotherapeutics on the expression of each signaling protein were quantified and are shown in Figure 5B–J (flavonoids) and Figure 5K–S (non-flavonoids). Like quercetin, fisetin, luteolin, and kaempferol dramatically reduced AKT phosphorylation compared to vehicle-treated cells (Figure 5B), without affecting total AKT protein expression (Figure 5C). All flavonoids, except senomorphic kaempferol, significantly suppressed PRAS40 phosphorylation (Figure 5D) without changes in total PRAS40 levels (Figure 5E). Interestingly, only quercetin and fisetin significantly reduced ERK1 (Figure 5F) and ERK2 phosphorylation (Figure 5H), without affecting total ERK1 (Figure 5G) or total ERK2 (Figure 5I) protein levels. Finally, all flavonoid senolytics (excluding kaempferol) significantly increased p53 expression, mirroring the effect of quercetin (Figure 5J).

Among the non-flavonoid senolytics, only dasatinib significantly altered these signaling mediators compared to vehicle treatment (Figure 5K–S). Specifically, dasatinib significantly reduced AKT phosphorylation (Figure 5K), without altering total AKT levels (Figure 5L). Although grouped non-flavonoid analysis showed a non-significant reduction in phospho-PRAS40 levels (Figure M), pairwise analysis revealed that dasatinib alone significantly decreased phospho-PRAS40 (*p* = 0.0078) without altering total PRAS40 levels. None of the non-flavonoids altered phospho-ERK1 (Figure 5O), total ERK1 (Figure 5P), phospho-ERK2 (Figure 5Q), or total ERK2 (Figure 5R) levels. Dasatinib was the only non-flavonoid that significantly enhanced p53 levels when compared to vehicle-eSCs (Figure 5S). The BH3 mimetics ABT-737 and navitoclax did not alter the expression of any of these signaling proteins (Figure 5K–S). Full blot images and band densities are provided in Appendix A, respectively. Similar signaling responses were observed in eSCs from endometriosis patients (Supplemental Data S3).

### 3.6. Elimination of Senescent Endometrial Stromal Cells (eSCs) by Senolytics

Senescence was investigated using fluorescent NanoJaggs, which are preferentially taken up by senescent cells [49]. First, we showed that NanoJaggs selectively accumulated in PALB-induced senescent eSCs when compared to vehicle-treated control eSCs (which have fewer senescent cells) (Figure 6A,B), though uptake varied across donor cells. Palbociclib was chosen to generate a nearly complete senescent cell population. Next, we showed that pre-treatment with the senolytic quercetin significantly reduced NanoJagg signals in both vehicle- and PALB-treated (senescent) cells (Figure 6C,D, respectively), suggesting its senolytic activity. Note that PALB-treated cells showed higher NanoJagg accumulation. To confirm that PALB-treated eSCs were indeed senescent, we sorted NanoJagg-positive and NanoJagg-negative eSCs using PALB- and vehicle-treated eSCs and assessed senescence biomarker expression. NanoJagg-positive (senescent) eSCs expressed higher p16 and phospho-Rb levels (markers of senescence) than NanoJagg-negative eSCs, including low passage vehicle-treated eSCs (Figure 6E, lanes 1–5), demonstrating loss of phospho-Rb (cell cycle arrest) and higher p16 levels and implementation of senescence. Also, expression of these senescence biomarkers was reduced following quercetin treatment (Figure 6E, lanes 6–9), reflecting senolytic elimination of p16^+^-senescent eSCs. Accordingly, the proliferation of PALB-treated cells was dramatically lower than that of vehicle-treated cells (Figure 6F,G), indicative of senescence-associated growth inhibition.

After verifying that NanoJagg accumulation by eSCs reflected senescence, we analyzed the effects of flavonoid and non-flavonoid senotherapeutics on NanoJagg accumulation by control-eSCs. As shown in Figure 6H, senolytic flavonoids (excluding the senomorphic kaempferol) significantly reduced NanoJagg uptake by control-eSCs under basal conditions, whereas all flavonoids tested markedly decreased NanoJagg accumulation (Figure 6I), reflecting the elimination of senescent cells. Among the non-flavonoids, only dasatinib significantly reduced NanoJagg-positive cells under basal (non-PALB) conditions (Figure 6J). In contrast, all non-flavonoids significantly reduced NanoJagg accumulation in PALB-induced senescent cells, with dasatinib demonstrating the most pronounced effect (Figure 6K), reflecting senolysis.

## 4. Discussion

We investigated a panel of senotherapeutic agents (Table 1) for their effects on endometrial stromal cell (eSC) biology using cells derived from menstrual effluent (ME). Specifically, this panel included both flavonoid and non-flavonoid senolytics, which selectively eliminate senescent cells, as well as a flavonoid senomorphic, which modulates senescence-associated secretory phenotype (SASP) and associated inflammation. The rationale for these experiments stemmed from prior rodent studies revealing that quercetin, a common flavonoid with senolytic and anti-inflammatory properties, reduced experimental endometriosis [28,29]. Moreover, quercetin has been shown to enhance the decidualization defect in eSCs obtained from endometriosis patients, often restoring function to levels observed in healthy controls [31]. Building on these findings, we now report that several common flavonoid senolytics, quercetin, fisetin, and luteolin, as well as the senomorphic kaempferol, consistently improved eSC decidualization capacity and significantly inhibited cell migration. In addition, dasatinib, a non-flavonoid senolytic, produced similar beneficial effects. These functional changes were accompanied by dramatic reductions in AKT phosphorylation and elevated p53 expression, along with evidence of their senolytic activity. In contrast, the BH3 mimetics ABT-737 and navitoclax were cytotoxic and did not confer functional benefits to eSCs.

Currently, there is a lack of reliably effective pharmacologic treatments for patients with endometriosis, a chronic condition affecting reproductive-age teens and women. Thus, our overall goal was to identify senotherapeutics that are suitable for evaluation in future clinical studies. Flavonoids, naturally occurring compounds found in fruits and vegetables, as well as other plant-based foods and supplements, are known for their potent antioxidant and anti-inflammatory properties, and several have demonstrated senolytic and/or senomorphic activities. Quercetin is one of the most well-studied flavonoids. It has been tested in numerous pre-clinical models and clinical trials for rheumatoid arthritis, diabetes, COVID-19, and insulin sensitivity in patients with PCOS [54]. Two major subclasses of flavonoids are flavonols (e.g., quercetin, fisetin, and kaempferol) and flavones (e.g., luteolin). While both subclasses exert anti-inflammatory effects, they differ by the presence of a hydroxyl group at C3 in flavonols that is absent in flavones [55].

In our study, the flavonoid senolytics, quercetin, fisetin, and luteolin, and the senomorphic, kaempferol, significantly enhanced eSC decidualization (Figure 1A–D), consistent with previous findings for quercetin at similar doses [30,31]. Decidualization, the differentiation of eSCs into secretory decidual cells, is essential for embryo implantation and placental development in humans and the few animals that menstruate [18]. Kaempferol, considered a phytoprogesterone, has been shown to induce progesterone-regulated gene expression in the mouse uterus [56]. However, to our knowledge this is the first study to assess the deciduogenic potential of kaempferol, fisetin, or luteolin in human eSCs.

Interestingly, among the non-flavonoids tested, dasatinib, a non-specific tyrosine kinase inhibitor, significantly increased decidualization, as indicated by increased IGFBP1 protein production across all doses tested (Figure 1F,H). This is consistent with a prior study showing increased *IGFBP1* mRNA expression by eSCs treated with dasatinib at similar doses (100 nM) [30]. Surprisingly, other non-flavonoids, ABT-737 and navitoclax, impaired decidualization at the tested doses (Figure 1E and Figure 1G, respectively), likely due to their cytotoxic effects. To our knowledge, this is the first report demonstrating their inhibitory effects on eSC decidualization. Notably, while dasatinib at 100 nM was mildly cytotoxic, it still significantly enhanced decidualization (Figure 1F). Defective decidualization has been reported using eSCs from patients with endometriosis [19,21,22,23], polycystic ovary syndrome (PCOS) [57], and conditions associated with infertility, as well as pre-eclampsia, recurrent pregnancy loss, and recurrent implantation failure [20,58]. Given its central role in reproductive success, impaired decidualization may represent a common pathological mechanism in these disorders, and thus may be a promising therapeutic target.

The lack of spontaneous decidualization in most laboratory animals is a barrier to studying and targeting this process in vivo. While most studies rely on endometrial biopsy-derived eSCs, we employed ME-derived eSCs. In addition to being non-invasive, ME-eSCs offer numerous advantages including convenient at home collection, the ability to obtain repeat or monthly samples, collection of phase-specific cell populations that are simultaneously being delivered to the peritoneal cavity, and the recovery of large volumes providing abundant eSCs. Notably, for clinical studies, ME-eSCs can be collected pre- and post-treatments or exposures, enabling ex vivo assessment of eSC responses to in vivo uterine conditions.

While all flavonoid senotherapeutics tested significantly inhibited eSC migration, kaempferol, the only non-senolytic senomorphic tested, was the least effective at the dose tested (Figure 4D). Similar to the decidualization results, dasatinib was the only non-flavonoid that significantly reduced eSC migration (Figure 4E). While enhanced stromal cell migration and invasiveness are implicated in the pathogenesis of endometriosis [59,60], no prior studies compared the effects of flavonoid and non-flavonoid senotherapeutics on eSC migration. This behavior may be partially driven by elevated estrogen levels commonly observed in endometriosis, which have been shown to enhance eSC motility [61]. Strategies to target the invasive and migratory capacity of eSCs have been proposed for treating endometriosis [62,63], as this approach may reduce lesion development and progression in the long-term.

To our knowledge, these studies are the first to demonstrate the utility of NanoJaggs for detecting cellular senescence in primary eSCs and for quantifying senolytic activity of various senotherapeutics using eSCs. Our findings show that flavonoid senolytics and dasatinib eliminate senescent cells (Figure 6H–K) and specifically, p16^+^-senescent-eSCs (Figure 6E). These findings support the hypothesis that these agents promote senolysis, thereby potentially enhancing the decidualization capacity of neighboring non-senescent eSCs.

Based on our prior studies, which identified elevated levels of senescence biomarkers in ME-eSCs from individuals with endometriosis [9], we previously proposed a model in which senescent eSCs compromise endometrial function and contribute to endometriosis lesion formation and disease progression by promoting inflammation and extracellular matrix remodeling leading to fibrosis. These findings align with other reports indicating that cellular senescence negatively impacts eSC function [8,64] and that decidual senescence “governs endometrial rejuvenation and remodeling at embryo implantation”, a process critical for fertility [65], which is often compromised in endometriosis. Thus, selectively targeting senescent cells with senotherapeutics may offer a promising strategy to improve endometrial function, mitigate endometriosis progression, and reduce associated infertility. While senescence pathways are incompletely understood and likely diverse depending on the tissue, cell type, and stimuli, they often target survival or anti-apoptosis mechanisms and related pathways, including those regulated by the p53/p21^CIP1/WAF1^ axis [66]. Interestingly, the p53 pathway and upstream pathways (AKT, ERK1/2) have been implicated in cell survival, decidualization, and migration. Only quercetin and fisetin reduced ERK1/2 phosphorylation (Figure 5F,H), which is consistent with its role in cell survival [67,68].

Flavonoids (quercetin, fisetin, kaempferol, and luteolin) and dasatinib had the most striking effects on reducing AKT phosphorylation in eSCs (Figure 5B,K) implicated in cell survival. scRNAseq data of vehicle- vs. quercetin-treated eSCs identified numerous genes implicated in regulating cell survival, either directly or indirectly (Figure 3 and Appendix A). For example, quercetin treatment downregulated *CITED2*, a gene whose loss has been linked to reduced AKT phosphorylation and subsequent modulation of MDM2-p53 interactions, resulting in p53 activation [69]. Numerous studies report that quercetin attenuates PI3K/AKT signaling, promotes apoptosis, inhibits proliferation, and reduces survival of eSCs and other cell types [29,31,70,71]. However, to our knowledge, these effects have not been systematically evaluated for non-quercetin flavonoid senotherapeutics or for dasatinib alone. In our study, both dasatinib and flavonoid senolytics consistently elevated p53 levels (Figure 5J,S), further supporting their role in modulating senescence-associated signaling pathways.

Uterine-specific deletion of *TP53* (eliminating p53 protein expression) in mice leads to premature senescence in decidual cells and pre-term birth, with marked increases in p-AKT and p21, reflecting a critical role for p53 in maintaining decidual cell homeostasis [72]. Accordingly, we observed that treating eSCs with flavonoid senolytics (quercetin, fisetin, and luteolin) and dasatinib enhance decidualization (Figure 1), reduced p-AKT and increased p53 (Figure 4), and eliminated senescent eSCs (Figure 6). Given the differences between mouse and human reproductive biology, human primary eSCs could be further studied to investigate the mechanisms of senescence in eSCs and decidual cells and their effects on fertility and pregnancy outcomes.

A favorable safety profile is critical when developing novel treatments for adolescents and young women of reproductive age with chronic conditions such as endometriosis. In this regard, our findings are encouraging. As expected for plant-derived compounds, the flavonoids quercetin, fisetin, kaempferol, and luteolin exhibited no cytotoxicity under decidualization conditions at 25 µM. Interestingly, fisetin and kaempferol (at 25 and 50 µM), as well as luteolin (at 25 µM only) showed cytoprotective effects (Figure 2A–D), possibly reflecting the clearance of harmful senescent cells (as supported by Figure 6H,I) and subsequent improvement in neighboring eSCs. Importantly, quercetin, fisetin and kaempferol are classified as generally recognized as safe (GRAS) by the FDA. In contrast, the non-flavonoid senolytics tested, ABT-737, navitoclax, and dasatinib, exhibited cytotoxicity, although dasatinib, which is not a BH3 mimetic, was the least cytotoxic and still improved decidualization (Figure 2H). Dasatinib is not recommended during pregnancy due to known fetal risks. ABT-737 has not been tested clinically due to delivery challenges and safety concerns, while navitoclax, its orally bioavailable analog, has shown promise in treating hematologic malignancies but carries the risk of thrombocytopenia [73].

Taken together, the natural flavonoid senolytics, quercetin, fisetin, and luteolin, demonstrated a favorable safety profile, consistently improved decidualization, reduced eSC migration, eliminated senescent cells, and lacked cytotoxic effects. These attributes provide a clear safety advantage over the non-flavonoids tested. Additionally, in a mouse model of idiopathic pulmonary fibrosis, quercetin promoted ligand-induced apoptosis in senescent fibroblasts and reduce fibrosis [71], suggesting that it may similarly target fibrosis that is often observed in endometriosis lesions [74]. Clinical trials are needed to determine the efficacy of quercetin and other flavonoid senolytics in the treatment of endometriosis.

While flavonoids offer promising safety and efficacy, their clinical translation has been limited by poor bioavailability. Numerous strategies have been employed to overcome this limitation, including nano-emulsions, liposomes, or other advanced delivery systems. One such example is Quercetin Phytosome^®^, which incorporates quercetin into a lecithin-based matrix that prevents self-aggregation and enhances absorption, leading a 20-fold increase in circulating levels compared to unmodified quercetin [75]. Thus, the bioavailability of flavonoids can be dramatically improved, and this will enhance therapeutic development and future application. Furthermore, localized drug delivery via vaginal or intrauterine administration should also be explored as a targeted approach for endometrial conditions.

Finally, in this study, participant eSCs showed marked between-subject variability in their response to flavonoid treatment across multiple assessments (e.g., decidualization, cytotoxicity, protein signaling expression, and cell migration). This variability likely stems from numerous factors, including participant age and uterine health status, genetic background, co-morbidities, and environmental exposures—differences that likely contribute to the numbers of pre-senescent and/or senescent endometrial cells and local levels of senescence-modulating or senescence-inducing factors. To develop more personalized flavonoid-based treatments for endometriosis and other uterine health disorders, future research will need to identify and characterize the specific factors responsible for these individual differences in treatment responses.

## 5. Conclusions

The lack of effective and well-tolerated non-hormonal pharmacotherapies remains a critical unmet need for patients with endometriosis. Hallmark features of endometriosis, such as impaired decidualization, accumulation of senescent eSCs, and enhanced migration and invasiveness of eSCs represent promising therapeutic targets. In this study, we found that flavonoid senolytics, quercetin, fisetin, and luteolin, robustly improved eSC decidualization, dramatically reduced eSC migration, and effectively eliminated senescent eSCs, without inducing cytotoxicity. These findings suggest that flavonoid-based senolytics may offer a safe, long-term, non-hormonal treatment strategy for patients with endometriosis.

## Figures and Tables

**Figure 1 biomolecules-15-00873-f001:**
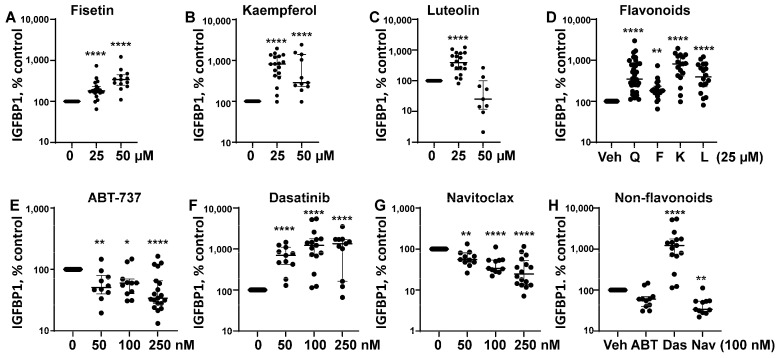
Flavonoid and non-flavonoid senotherapeutics differentially affect human endometrial stromal cell (eSC) decidualization. (**A**–**D**) Human eSCs were treated with flavonoids: fisetin (**A**), kaempferol (**B**), and luteolin (**C**) at 0–50 µM (0 = vehicle) or (**D**) all flavonoids (quercetin, Q; fisetin, F; kaempferol, K; luteolin, L) at 25 µM vs. vehicle (Veh) for 4 h prior to cAMP + MPA. Decidualization was determined by measuring IGFBP1 production in the culture supernatants 48 h later by ELISA. (**E**–**H**) eSCs were treated with non-flavonoids: ABT-737 (**E**), dasatinib (**F**), and navitoclax (**G**) at 0–250 nM (0 = vehicle) or all non-flavonoids (ABT-737, ABT; dasatinib, Das; navitoclax, Nav) at 100 nM vs. vehicle (Veh) (**H**) for 4 h prior to decidualization and assessment of IGFBP1, as described above. Data are shown % control IGFBP1 values with vehicle-treated cells = 100%. Each point represents mean data from one individual’s eSCs, with median and IQR shown for each group. * *p* < 0.05 vs. Veh; ** *p* < 0.01 vs. Veh; **** *p* < 0.0001 vs. Veh.

**Figure 2 biomolecules-15-00873-f002:**
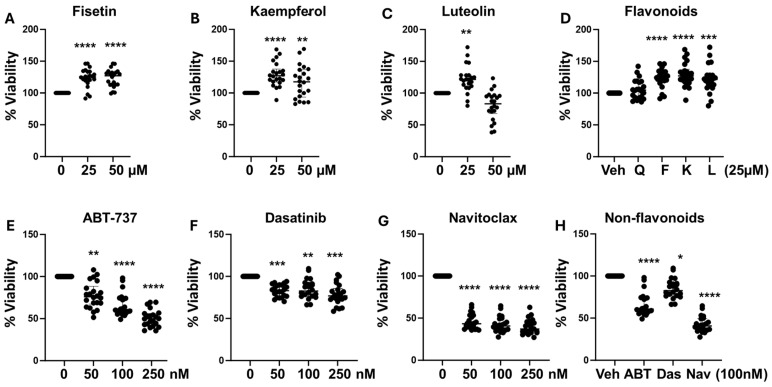
Flavonoid senotherapeutics are mostly non-cytotoxic compared to non-flavonoid senotherapeutics. (**A**–**D**) Human eSCs were treated with flavonoids: fisetin (**A**), kaempferol (**B**), and luteolin (**C**) at 0–50 µM (0 = vehicle) or (**D**) all flavonoids at 25 µM (quercetin, Q; fisetin, F; kaempferol, K; luteolin, L) vs. vehicle (Veh) for 48 h and assessed for cytotoxicity using the neutral red assay. (**E**–**H**) Human eSCs were treated with non-flavonoids, ABT-737 (**E**), dasatinib (**F**), and navitoclax (**G**) at 0–250 nM (0 = vehicle) or all non-flavonoids at 100 nM (**H**) (ABT-737, ABT; dasatinib (Das); navitoclax (Nav) vs. vehicle (Veh) and assessed for cytotoxicity by neutral red. Data are shown as % control viability values with vehicle-treated cells = 100%. Each point represents mean data from one individual’s eSCs, with median and IQR shown for each group.* *p* < 0.05, ** *p* < 0.01 vs. Veh; *** *p* < 0.001 vs. Veh **** *p* < 0.0001 vs. Veh.

**Figure 3 biomolecules-15-00873-f003:**
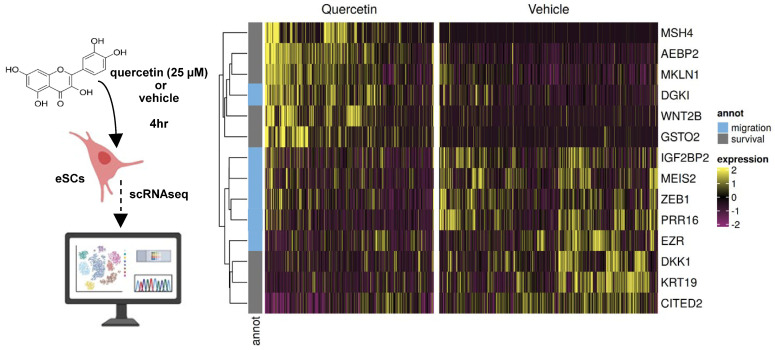
Human eSCs (n = 4 healthy controls) were treated with quercetin (25 µM) vs. vehicle for 4 h and then analyzed by scRNAseq. Heat map shows differentially expressed genes implicated in cell migration and survival.

**Figure 4 biomolecules-15-00873-f004:**
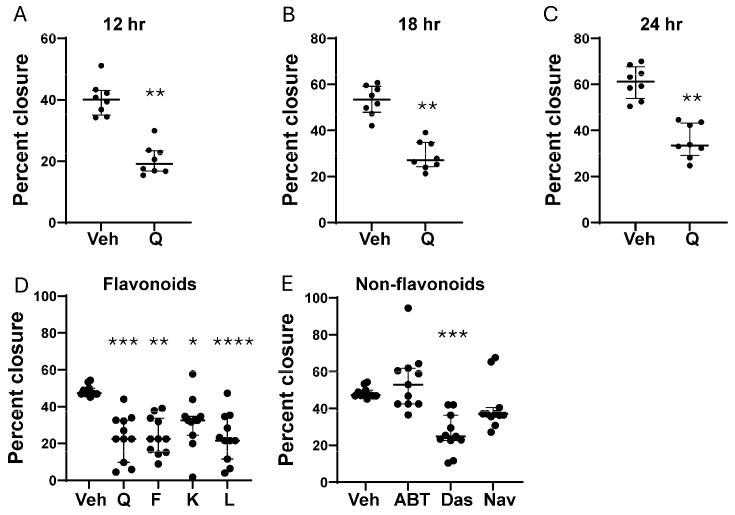
The effects of flavonoid and non-flavonoid senotherapeutics on eSC migration. (**A**–**C**) Confluent eSCs were treated with vehicle (Veh) or quercetin (Q) at 25 µM and then analyzed for migration using the wound closure assay 12 h (**A**), 18 h (**B**), and 24 h (**C**) post-scratch (n = 8 subjects). Cell migration was assessed as percentage wound closure at each time point. (**D**) Confluent eSCs (n = 11 subjects) were treated with flavonoids: quercetin (Q), fisetin (F), kaempferol (K), and luteolin (L) at 25 µM vs. vehicle (Veh). (**E**) eSCs (n = 11 subjects) were treated with vehicle or non-flavonoids (ABT-737 (ABT), dasatinib (Das), navitoclax (Nav)) at 100 nM and wound closure was compared to vehicle (Veh)-treated cells. (**D**,**E**) Data are shown as percentage closure measured at the time when vehicle-treated cells showed approximately 50% closure. Each point represents mean data from one individual’s eSCs, with median with IQR shown for each group. * *p* < 0.05 vs. Veh; ** *p* < 0.01 vs. Veh; *** *p* < 0.001 vs. Veh **** *p* < 0.0001 vs. Veh.

**Figure 5 biomolecules-15-00873-f005:**
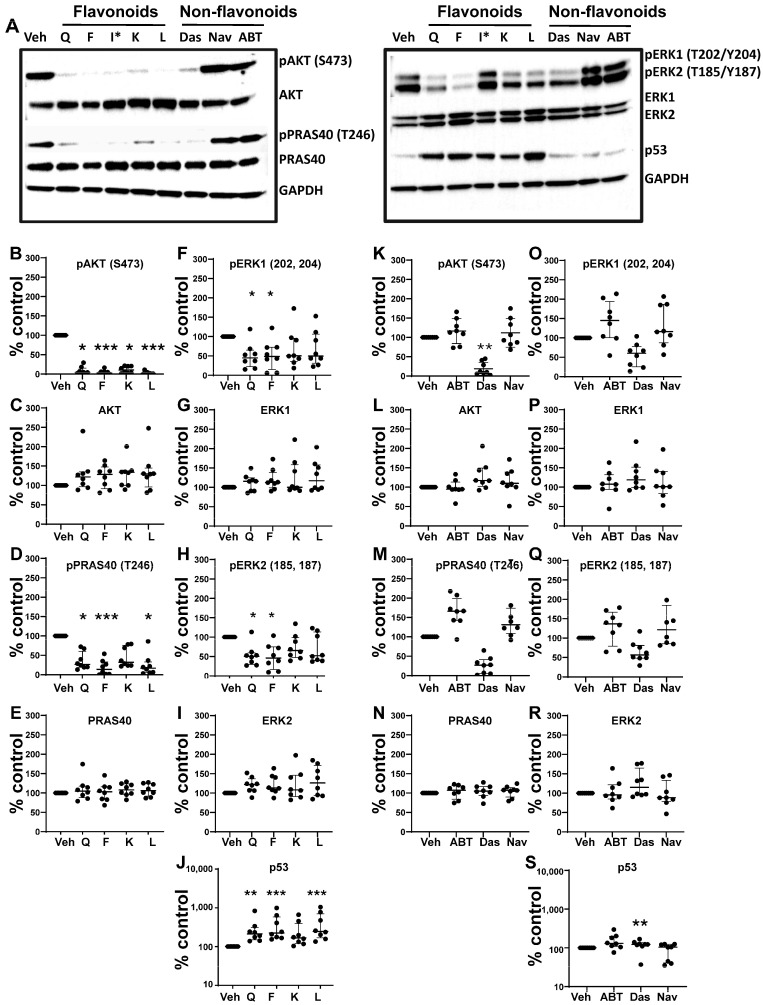
The differential effects of flavonoid and non-flavonoid senotherapeutics on eSC signaling pathways. (**A**) Representative western blot images for one control participant’s eSCs treated with vehicle (Veh) vs. senotherapeutics, flavonoids: quercetin (Q), fisetin (F), kaempferol (K), and luteolin (L) at 25 µM and non-flavonoids: ABT-737 (ABT), dasatinib (Das), navitoclax (Nav)) at 100 nM. Note: I* was not included in the analyses. (**B**–**J**) eSCs (n = 8 subjects) were treated with vehicle (Veh) or flavonoids, including quercetin (Q), fisetin (F), kaempferol (K), or luteolin (L) at 25 µM for 4 h before analyzing cell lysates by western blotting for p-AKT (**B**), total AKT (**C**), p-PRAS40 (**D**), total PRAS40 (**E**), p-ERK1 (**F**), total ERK1 (**G**), p-ERK2 (**H**), total ERK2 (**I**), and p53 (**J**). (**K**–**S**) eSCs (from the same 8 subjects) were treated with vehicle (Veh) or various flavonoids, including ABT-737 (ABT), dasatinib (Das), or navitoclax (Nav) for 4 h before analyzing cell lysates for the same proteins as above. Band densities were normalized to GAPDH and corrected for total non-phosphorylated proteins, where appropriate, and shown as % control between Veh- vs. flavonoid-treated eSCs (**B**–**J**) or Veh- vs. non-flavonoids (**K**–**S**), with vehicle-treated cell values set to 100%. Each point represents mean data from one individual’s eSCs (from 8 subjects), with median and IQR shown for each group. * *p* < 0.05; ** *p* < 0.01; *** *p* < 0.001. All uncropped blot images and band densities (with calculations) are shown in Appendix A, respectively.

**Figure 6 biomolecules-15-00873-f006:**
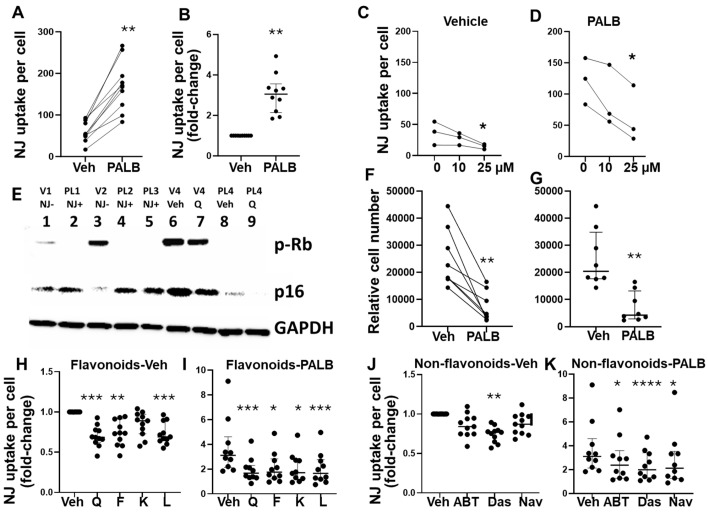
Senotherapeutics effectively eliminate senescent cells. (**A**,**B**) Vehicle-eSCs vs. palbociclib (PALB)-treated eSCs (to induce senescence) analyzed for NanoJagg (NJ) accumulation (n = 10 subjects), with data shown as NJ uptake per cell as mean paired data connected by a line for each participants’ eSCs (**A**) and NJ uptake per cell as fold-change, where vehicle-treated eSCs = 1; each point represents mean data from one individual’s eSCs, with median and IQR shown for each group (**B**). (**C**,**D**) Vehicle-treated eSCs (**C**) or PALB-treated (senescent) eSCs (**D**) treated with vehicle or quercetin (at 10 or 25 µM) and assessed for NJ uptake (3 subjects each). Data are shown as mean paired data connected by a line. (**E**) Western blot showing NJ^−positive^ (NJ+) vs. NJ^−negative^ (NJ-) sorted eSCs (lanes 1–5) and non-sorted eSCs ± quercetin (Q, 25 µM) (lanes 6–9). Lanes 1–5: paired NJ^+^ vs. NJ^−^ following vehicle-(V) or palbociclib (PL)-treated eSCs from subject 1 (V1 and PL1), subject 2 (V2 and PL2) and subject 3 (PL3) blotted with antibodies to senescence markers (p-Rb and p16) and GAPDH. Note V2 eSCs are low passage (p2) and other eSCs are p6–p8. Lanes 6–9: eSCs from subject 4 (p6–p8) were pre-treated with vehicle (V4) vs. PALB (PL4) to induce senescence followed by vehicle (Veh) vs. quercetin (Q, 25 µM) and then blotted with antibodies to senescence markers (p-Rb and p16) and GAPDH. Full blots and band densities are in Appendix A. (**F**,**G**) Comparison of vehicle-eSCs vs. PALB-eSCs for cell proliferation (n = 8 subjects); F shows paired data (relative cell number) connected by a line for each participants’ eSCs and G shows proliferation data (relative cell number) where each point represents mean data from one individual’s eSCs, with median with IQR shown for each group. (**H**–**K**) eSCs (from 11 subjects) were either pre-treated with vehicle (Veh) (**H**,**J**) or PALB (**I**,**K**) to induce senescence before treatment with flavonoids (**H**,**I**) or non-flavonoids (**J**,**K**) and then assessed for NJ accumulation (shown as NJ uptake per cell as fold-change as in (**B**)). Each point represents mean data from one individual’s eSCs, with median and IQR shown for each group. * *p* < 0.05; ** *p* < 0.01; *** *p*  <  0.001; **** *p* < 0.0001.

**Table 1 biomolecules-15-00873-t001:** Panel of senotherapeutic agents tested.

Senotherapeutic	Flavonoid Y/N Subclass	Senescence-Related Function	Doses Used	Other Functions
**Quercetin** 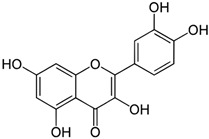	Yes Flavonol	Senolytic Senomorphic	25–50 µM	Antioxidant anti-inflammatory inhibits CDKs and cyclins Kinase inhibitor (PI3K, AKT)
**Fisetin** 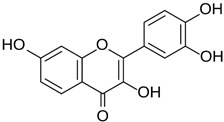	Yes Flavonol	Senolytic Senomorphic	25–50 µM	Antioxidant anti-inflammatory inhibits CDK6 Kinase inhibitor (PI3K, AKT)
**Kaempferol** 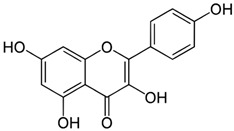	Yes Flavonol	Senomorphic	25–50 µM	Antioxidant anti-inflammatory; targets SASP Kinase inhibitor (AKT)
**Luteolin** 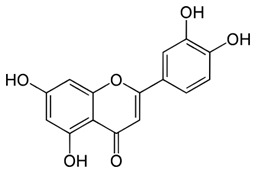	Yes Flavone	Senolytic (weak) Senomorphic	25–50 µM	Anti-inflammatory inhibits cyclins Kinase inhibitor (p38)
**ABT-737** 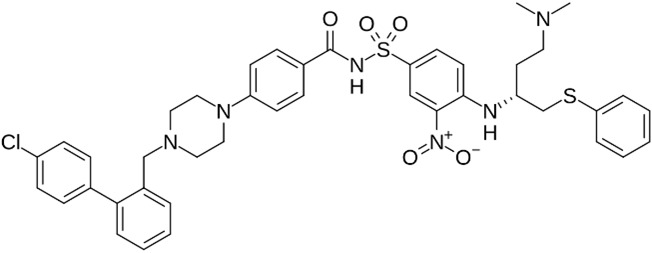	No BH3 mimetic	Senolytic	50–250 nM	Inhibits Bcl-2 family members (Bcl-2 and Bcl-xl)
**Dasatinib** 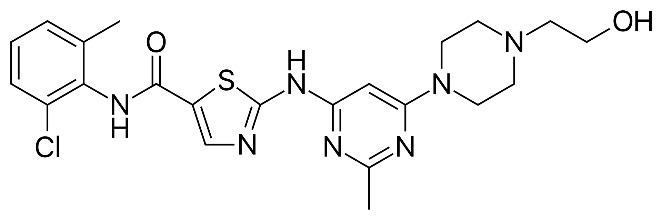	No Tyrosine kinase inhibitor	Senolytic	50–250 nM	Kinase inhibitor (Src/Abl) Anti-inflammatory (indirectly)
**Navitoclax** **(ABT-263)** 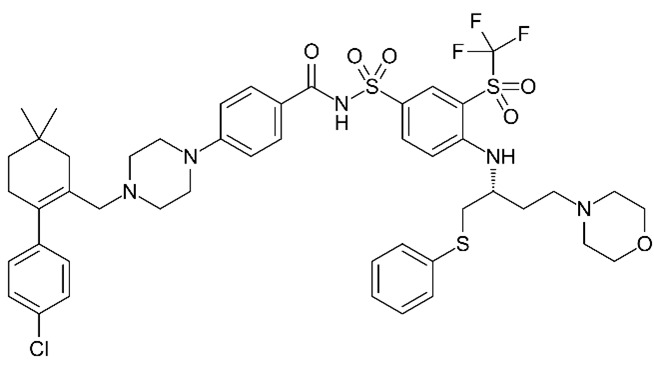	No BH3 mimetic	Senolytic	50–250 nM	Inhibits Bcl-2 family members (Bcl-2 and Bcl-xl)

## Data Availability

The datasets analyzed during this study are available from the corresponding authors upon reasonable request. Materials described in the manuscript are commercially available, with the exception of the primary menstrual effluent-derived endometrial stromal cells (eSCs) and the NanoJagg probes.

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
