# Peer review of "Targeting Cellular Senescence to Enhance Human Endometrial Stromal Cell Decidualization and Inhibit Their Migration"

_biomolecules, 2025, doi:10.3390/biom15060873_

Round 1

Reviewer 1 Report

Comments and Suggestions for Authors

It is an interesting manuscript, and the work is well done. The introduction section can be enriched and written more clearly for readers unfamiliar with the subject.

Author Response

3. Point-by-point response to Comments and Suggestions for Authors

Comment 1: The introduction section can be enriched and written more clearly for readers unfamiliar with the subject.

Response: Thank you for pointing out that our introduction could be improved. We agree with your comment, as readers may not completely understand the significance of this research without better contextualization. Therefore, we have enhanced the introduction to better orient the readers to the topics of senescence and endometriosis. See revised introduction on pages 4-6 of the new manuscript. In the revised text we better described senescence (particularly with respect to uterine health), the potential role of senescence in endometriosis and related fertility and why targeting senescence in the setting of endometriosis may be beneficial.

Reviewer 2 Report

Comments and Suggestions for Authors

The manuscript by Delenko et al. provides valuable insights into the negative impact of senescent cells on endometrial stromal cell function and their role in the formation and progression of endometriotic lesions. This knowledge is essential for the development of targeted therapies for endometriosis, but there are some issues that need to be addressed before final acceptance.

MAJOR

  1. One major concern is the lack of consideration of individual variability in patient response to flavonoid treatment. Factors such as genetics, disease severity, and co-morbidities can influence treatment outcomes and necessitate a personalized approach. It would be beneficial to discuss these issues in the Discussion section of the manuscript.
  2. Please provide information on how validation of in vitro decidualization was conducted. Subchapter 2.3 does not contain sufficient information.

MINOR

  1. Table 1 - Several formulas are not visible clearly.
  2. Please indicate the nature of the control vehicle.

Author Response

Response to Reviewer 2 Comments

1. Summary      The manuscript by Delenko et al. provides valuable insights into the negative impact of senescent cells on endometrial stromal cell function and their role in the formation and progression of endometriotic lesions. This knowledge is essential for the development of targeted therapies for endometriosis, but there are some issues that need to be addressed before final acceptance.

Response: Thank you for taking the time to review our manuscript. We have addressed your concerns. Please see the responses below and the corresponding revisions/corrections highlighted/in track changes in the re-submitted documents and files.

2. Questions for General Evaluation

Reviewer’s Evaluation

Response and Revisions

Does the introduction provide sufficient background and include all relevant references?

Yes-okay for publication

Are all the cited references relevant to the research?

Yes-okay for publication

Is the research design appropriate?

Yes-okay for publication

Are the methods adequately described?

Can be improved

The methods section has been revised, as suggested

Are the results clearly presented?

Yes-okay for publication

Are the conclusions supported by the results?

Yes-okay for publication

Are all tables and figures clear  

Can be improved

Table 1 has been improved to better show images

1.     Summary: The manuscript by Delenko et al. provides valuable insights into the negative impact of senescent cells on endometrial stromal cell function and their role in the formation and progression of endometriotic lesions. This knowledge is essential for the development of targeted therapies for endometriosis, but there are some issues that need to be addressed before final acceptance.

Response: Again, thank you for the thoughtful and thorough review of our manuscript. We have addressed your concerns below and in the revised manuscript. Details are below.

MAJOR

  1. One major concern is the lack of consideration of individual variability in patient responses to flavonoid treatment. Factors such as genetics, disease severity, and co-morbidities can influence treatment outcomes and necessitate a personalized approach. It would be beneficial to discuss these issues in the Discussion section of the manuscript.

Response 1: Thank you for pointing out our lack of consideration of individual variability in the responses of participants’ stromal cells to flavonoid treatment. We agree it would be beneficial to discuss this variability. We have significantly revised the discussion section to address the variability in eSCs from various donors. See revised discussion section on page 27 and below (just before the conclusions).

“Finally, in this study participant eSCs showed marked ‘between subject’ variability in their response to flavonoid treatment across multiple assessments (e.g., decidualization, cytotoxicity, protein signaling expression, and cell migration). This variability likely stems from numerous factors, including participant age and uterine health status, genetic background, co-morbidities, and environmental exposures - differences that likely contribute to the numbers of pre-senescent and/or senescent endometrial cells and local levels of senescence-modulating or senescence-inducing factors. To develop more personalized flavonoid-based treatments for endometriosis and other uterine health disorders, future research will need to identify and characterize the specific factors responsible for these individual differences in treatment responses.”

  1. Please provide information on how validation of how in vitro decidualization was conducted. Subchapter 2.3 does not contain sufficient information.

Response 2: We agree and have revised Subchapter 2.3 to provide more information about our validation assays/prior studies. See page 9 and below.

“Specifically, doses of deciduogenic agents and time points of assessment were based on our prior studies and previously published studies, allowing for the comparison of results across studies. Additionally, the 48 hr time point is consistent with our prior studies using ME-derived eSCs where the decidualization response is robust enough to detect both positive and negative effects and prior to its peak or completion [19, 21, 31]. Furthermore, the combination of deciduogenic stimuli (cAMP+MPA) was chosen because recent transcriptomic and cell-based studies suggest it best reflects decidualization in vivo [36]. Note: concentrations of flavonoids used were guided by our previous studies with quercetin [31] (using doses that inhibited proliferation and enhanced decidualization without cytototoxicity) and by published studies (quercetin [37-39], fisetin [40-42]; kaempferol [37, 43, 44]; and luteolin [45, 46]) that used similar concentrations. Concentrations for non-flavonoids were based a study using dasatinib-treated human cells (100nM) [47] and initial studies; doses for non-flavonoids were mainly below those described in the literature for primary cells due to cytotoxicity.”

MINOR

  1. Table 1 - Several formulas are not visible clearly.

Response 1-minor: We believe the images were too small and had low resolution in the original Table 1. We have revised Table 1 to cover two pages (instead of one) and have inserted larger and higher resolution images. Hopefully, this version of Table 1 is acceptable (if not, we will work with the editors to improve Table 1). See revised Table 1-now spread across 2 pages (pages 6-7).

  1. Please indicate the nature of the control vehicle.

Response 2-minor: We have revised the methods section to better define ‘vehicle’ (<0.1%DMSO in assay media). See page 8 and below. Additionally, we have indicated that 0 dose = vehicle in multiple figure legends. Note: as described in the Chemical and Reagents section, all senotherapeutics, including quercetin were prepared in DMSO, as directed by the manufacturer.

“The next day, cells were treated with either vehicle (assay media containing <0.1%DMSO [final], equivalent to the highest dose of DMSO used when treating with various agents), quercetin (25μM), or various senotherapeutics (flavonoids: 25-50μM; non-flavonoids: 50-250nM) (all prepared in DMSO, as described above) diluted in assay media for 4 hrs before the addition of cAMP (0.5 mM) + MPA (10-7 M) to induce decidualization (3-4 wells/condition), followed by measurement of IGFBP1 protein 48 hrs later, as previously described [19, 31]”.

Round 2

Reviewer 2 Report

Comments and Suggestions for Authors

Congratulations! You successfully improved the manuscript.